# Communicating the risk of contracting Zika virus to low income underserved pregnant Latinas: A clinic-based study

**Suhasini Ramisetty-Mikler**[1]*, **LeAnn Boyce**[2]

**1** Population Health Sciences, School of Public Health, Georgia State University, Atlanta, GA, United States of America, **2** Department of Information Science, University of North Texas, Denton, TX, United States of America

* sramisettymikler@gsu.edu

## Abstract

### Objective

Frequent travel between the Southern border states in the USA, Mexico, and Latin American countries increases the risk of the Zika virus (ZIKV) spread. Patient education on virus transmission is fundamental in decreasing the number of imported cases, particularly among pregnant women.

### Methods

The study used cross-sectional methodology to investigate information sources and knowledge concerning the ZIKV virus among 300 under-served pregnant Latinas recruited from prenatal care clinics in the North Texas region. Bivariate and multiple logistic regression models were used to investigate associations between the primary outcomes and patient characteristics.

### Results

Physicians, nurses, and families are the major sources for pregnancy information, while media/internet (65%) and physician/nurse (33%) are the main sources for ZIKV information. Less than one-half of the mothers reported that their physician/nurse did not discuss safe sexual practices or inquired about their sexual practices. A considerable proportion of women from the community clinic were neither warned nor queried about travel to ZIKV risk countries. There is an overall understanding of Zika virus transmission, symptoms, complications, and recommended guidelines. Younger age and single mother status are risk factors for lack of ZIKV knowledge. Foreign-born mothers are 2.5–3.0 times more likely to have knowledge on disease transmission, symptoms, and microcephaly condition. While, younger mothers (18–24) are less likely to have knowledge of ZIKV infection symptoms (fever, rash and pink eye) and transmission of infection via unprotected sexual (vaginal, anal, or oral) behavior, compared to older mothers.

**Data Availability Statement:** Data is available on the Open ICPSR repository: https://www.openicpsr.org/openicpsr/project/125702/version/V1/view (DOI: https://doi.org/10.3886/E125702V1).

**Funding:** This study was supported by University of North Texas faculty internal grant awarded to the first author. The funders had no role in study design, data collection and analysis, decision to publish, or preparation of the manuscript. None of the authors received salary under this specific funding.

**Competing interests:** The authors have declared that no competing interests exist.

## Conclusions

Interventions are needed to heighten the knowledge of ZIKV, particularly among women of reproductive age and their male partners in the community health care setting. Our study underscores the need for health care providers to be trained in delivering messages to enhance risk perception during health emergencies to vulnerable and underserved families (lower economic background, language ability, and culture). During health emergencies, clinics must disseminate crucial information via multi modalities to ensure messages reach the targeted patients.

## Introduction

The risk of contracting the Zika virus (ZIKV) infection during pregnancy is a public health concern because of its serious consequences on birth outcomes [1]. Patient education on virus transmission is fundamental for the reduction of imported cases, especially in pregnant women, as it can cause serious congenital abnormalities including microcephaly and a wide range of neurological problems in children. Research is sparse on the outcomes of educational programs targeting pregnant women, in particular, pregnant women in the border states (e.g. California, Texas, Florida), who are likely to visit their families living in the known regions of ZIKV outbreak. It is imperative to investigate whether the crucial information concerning ZIKV is delivered to them and the channels through which women access information. This is significant for the Hispanic population in border states who may frequently travel to Mexico and other Latin American countries for family visits.

Following the ZIKV outbreak in Summer of 2015, approximately 216.3 million from ZIKV outbreak areas traveled (via air, by road, or by water) to the United States by the end of March 2016. Texas and Florida are the two border states with the largest population of women of childbearing age at an estimated 51.7 million and an estimated 2.3 million pregnant women [2, 3]. According to the ArboNET, during the three years following the ZIKV outbreak (Dec-2015 to March-2018), there were 116 live births with Zika-associated congenital abnormalities, and nine fetal deaths and pregnancy losses [4] contributed mostly by California, Florida, and Texas. A total of 588 travel associated (Mexico/Central America) cases, including 139 pregnant women in California and 315 travel-related cases, all except for a few local transmissions in Texas were reported.

Following the outbreak, the Centers for Disease Control and Prevention (CDC) issued guidelines on primary mitigation strategies, including travel restriction to outbreak areas, in particular, for pregnant women. Despite the media coverage of ZIKV cases in the United States and around the world, 24% of Americans reported knowing only a little about ZIKV, and 15% said they have not heard anything. Those who are aware are not particularly worried about its potential negative effects [5]. It is imperative to investigate whether the crucial information concerning the ZIKV is delivered through credible sources to the vulnerable populations during public health emergencies. The primary defense against the spread of the virus is to increase awareness through public education and information access to the entire population.

In general, studies have shown prenatal providers are the most trusted sources of health information, however, there is little knowledge on what other communication modalities and sources pregnant women prefer during an evolving public health emergency like ZIKV. Research on awareness and information-seeking behavior among the underserved Latino

population is sparse [6], specifically, with regards to imparting knowledge on the threat of ZIKV to Latino population in the border states. Two studies conducted on women in reproductive age [7] and pregnant women [8, 9] in Miami, Florida, reported that generally women have knowledge about the virus, its transmission, and prevention. In a North Carolina study of pregnant and not pregnant Latina, Zhou-Talbert [10] found that most participants had moderate/high knowledge of the virus transmission (92.5%) and symptoms (73.2%). However, they had limited knowledge of preventing the disease via sexual contact and congenital transmission. With regard to information sources, Ellingson, Bonk, and Chamberlain [11] reported that the most popular resource for obtaining ZIKV information among pregnant women was the CDC website (73.0%). Further, compared to African American women, Hispanic women (nearly 4 times) and White women (2 times) were significantly more likely to discuss about ZIKV with their providers. Studies also revealed that practicing healthcare providers need to improve communication strategies during public health emergencies [12].

Our study focuses on a unique vulnerable population in the state of Texas recruited from community-based healthcare clinics referred as Federally Qualified Health Centers (FQHC) that target low-income families. The FQHCs are funded by the Human Resources and Services Administration (HRSA) Health Center Program to provide comprehensive primary care services in underserved and rural areas identified by HRSA. FQHCs are required to accept all patients, regardless of their ability to pay or health insurance status. There are 73 FQHCs serving patients in Texas, operating in more than 300 sites [13]. One of the FQHCs in the North Texas Central Region located in the city of Denton (county seat of Denton County) is one of the three study clinic sites. The study sites which are located in the city of Denton are approximately 500 miles north of the Texas-Mexico border.

## Objective

There is a paucity of data on pregnant Latinas regarding their health information seeking behavior and health literacy. Underserved pregnant Latinas are at higher risk for low health literacy, particularly those who are non-native English speakers [14, 15]. The current study investigated clinic-based health risk education, information sources, knowledge of ZIKV and its correlates among underserved pregnant Latinas that are seldom researched. This study is one of the first to directly inquire about ZIKV related health literacy (knowledge and awareness) and information sources among low income, underserved, and undocumented pregnant Latinas in one of the border states. Specifically, the study examined (a) the mothers' knowledge on ZIKV transmission, symptoms, its impact on newborn, and prevention strategies, (b) information sources of ZIKV (mass media/family/physicians/clinics), and (c) modes of ZIKV information dissemination in the clinics. The study investigated two research questions:

1. Is there a difference in ZIKV knowledge based on the mother's demographics (country of birth, age, education, income, and marital status)? and

2. Is there a difference in the communication of health care providers (HCP) and information dissemination based on the clinic type (Federally funded versus Private)?

For this study, a Health Care Professional (HCP) refers to a physician, a registered nurse, a nurse practitioner, or clinic staff.

## Materials and methods

The study was approved by the University of North Texas institutional review board (IRB #16–317). The participants provided a written consent before completing the survey and the data were analyzed anonymously.

## Participants

Nearly 40% of Texas population is of Hispanic/Latino origin [16, 17] and in much higher concentrations between 70% and 96% in border counties (e.g. Brooks County, Cameron County, Webb County). Nearly 21% of Hispanics in Texas live in poverty compared to only 9% of Whites. Further, 37% of Hispanics are uninsured compared to 14% of Whites [18].

The study used self-reported survey data from pregnant Latinas (18+ years) that reside in Denton County which in 2019, had an estimated population of 887,207 where 19.5% are of Hispanic ethnic background. A total of 300 pregnant Latinas including 192 from a Federally Qualified Health Clinic (FQHC) and 108 from two private clinics that serve low-income families, completed the survey. The Health Services for North Texas is designated FQHC that serves the largest proportion (85%) of low-income Latino population in Denton County, which is part of the Dallas-Fort Worth Metroplex.

The inclusion criterion is of Hispanic origin regardless of their Spanish-speaking and or reading ability. If the mothers marked their ethnicity as Hispanic or Latin American origin in the intake form, the nursing staff recruited them into the study. Thus, all Latino women who visited the selected clinics (between October 2016 and February 2018) were solicited to take part in the study. The investigator, together with the office manager trained the nursing staff on the identification of mothers from medical intake forms, obtaining consent, recruitment, and survey procedures.

## Ethical approval

All procedures performed in this study involving human participants were in accordance with the ethical standards of the institutional and/or national research committee and with the 1964 Helsinki declaration and its later amendments. The study procedures, survey instrument, and patient consent form were approved by the University's Institutional Review Board, with which the authors are affiliated.

## Measures and data collection

All Latino women (self-identified as Hispanic or Latino on the intake form) who visited these three clinics (between October 2016 and February 2018) were solicited to take part in the study. During one of the mother's scheduled prenatal visits, a staff member explained the purpose of the study and obtained informed consent prior to completing the survey. Mothers completed the survey in the language of their choice (English or Spanish) while they waited in the front reception area. Once they returned the survey, the staff checked for its completeness and gave the mothers a $10/- Wal-Mart gift card for completing the survey. The administrative staff assigned a random number to each mother, maintained a log of those who completed the survey, time and date of completion, and who received the gift card. This process ensured that each mother took the survey only once. The staff also checked the surveys for their completeness and the quality of the responses. The actual data collection took nearly 12 months (January 2017-February 2018) to obtain the desired sample of 300 mothers. Out of 305 mothers who were approached, only five mothers refused which resulted in 98% response rate. On average, mothers took 21 minutes (SD = 9.8) to complete the survey.

The survey included questions specifically developed for the study to inquire about

a. awareness of the ZIKV and its impact on pregnancy and newborns. Five statements pertaining to the awareness of CDC recommendations concerning ZIKV transmission were asked with yes or no response options: i) Pregnant women should not travel to any area where ZIKV transmission is ongoing; ii) Travelers should take protective measures, including the

use of insect repellent, to prevent mosquito bites both during travel and for 3 weeks after returning to their home country. Such measures include wearing long-sleeved shirts and long pants; staying in places with air conditioning and window and door screens to keep mosquitoes outside; sleeping under a mosquito bed net, and using insect repellents with active ingredients (e.g., DEET); iii)Travelers should prevent possible sexual transmission after returning home (if travelled to ZIKV outbreak areas like by correctly using condoms every time they have sex or by abstaining from sex). Males should use condoms for at least 8 weeks after travel or, if symptomatic for Zika virus infection, for 6 months from the start of symptoms; iv) After returning from a country with Zika virus transmission, men with pregnant partners should use condoms or not have sex for the duration of the pregnancy; and v) Couples who want to try to get pregnant after attending the Olympic and Paralympic Games should wait at least 8 weeks, and 6 months if the male partner has symptomatic Zika virus infection. Survey questions regarding ZIKV awareness and prevention strategies were based on the information provided by CDC [19].

b. Ten knowledge of virus transmission, symptoms, and primary prevention strategies from contracting the ZIKV (e.g., mosquito bites, sexual transmission: Each knowledge area was asked in a question format with yes or no response option (see the list of ten questions in Table 1);

c. travel behavior: whether they and or their partner traveled to ZIKV outbreak areas 3 months prior to getting pregnant or planning to travel while being pregnant; and

d. information sources (mass media, physicians, clinics, and friends/family) that are likely to impact ZIKV literacy Mothers were asked to indicate various sources through which they learned about Zika virus. A list of possible sources (Physician/nurse, Information available at the clinic, Books/magazines, TV/Media/internet, Family, and) Friends) with check all that apply option was provided.

Specific questions on physician-patient communication and information dissemination included: Whether the physician/nurse specifically educated mothers about Zika virus and its effect on the babies voluntarily or information was given only after the mother asked for it; whether the information was provided in Spanish or in English; whether the clinic or hospital displayed printed information (posters on walls) or have handouts given in English and or Spanish. Mothers also reported on whether the nurse/physician warned against travelling to places where Zika virus is active; and whether the physician/nurse discussed about safe sexual practices (e.g., using a condom consistently or avoiding sex) during pregnancy if they or the partner are exposed to Zika virus.

The instrument was translated from English into Spanish and back translated to English. The final version was pilot tested (N = 20) with the clinic staff and graduate students for clarity and consistency.

## Study variables

Dependent variables in the study were ten questions about knowledge and behavior concerning the ZIKV designed based on information available at the CDC website. Each statement was asked in the question format with yes or no response options (see Table 1). For example, "Are you aware that the Zika virus is transmitted primarily by mosquito bites?" Mothers were also asked to report on their awareness of the CDC recommendations, particularly for pregnant women including travel ban to ZIKV threat areas, to avoid sexual contacts, use of a

**Table 1. Bivariate associations between demographic factors and knowledge statements (percent of "yes").**

| Demographic factors | Knowledge Statements[*] | | | | | | | | | |
|---|---|---|---|---|---|---|---|---|---|---|
| | **a** | **b** | **c** | **d** | **e** | **f** | **g** | **h** | **i** | **j** |
| **Country of Birth (Total)** | 92 | 79.9 | 65.6 | 65.2 | 66.9 | 79.6 | 97.3 | 62.8 | 71.8 | 57.9 |
| USA (n = 115) | 86.2 | 67.2 | 57.8 | 48.3 | 54.3 | 66.4 | 94.8 | 55.2 | 70.7 | 56.5 |
| Foreign born (n = 184) | 95.6 | 87.9 | 70.5 | 76 | 74.9 | 88 | 98.9 | 67.6 | 72.5 | 58.9 |
| $\chi 2$ (df = 1) | 8.54 | 18.83 | 5.1 | 23.98 | 13.54 | 20.4 | 4.43 | 4.67 | 0.12 | 0.16 |
| p-value | **0.003** | **0.001** | **0.02** | **0.001** | **0.001** | **0.001** | **0.03** | **0.03** | 0.73 | 0.69 |
| **Age (Total)** | 92 | 79.9 | 65.6 | 65.2 | 66.9 | 79.6 | 97.3 | 62.8 | 71.8 | 57.9 |
| 18–24 yrs (n = 104) | 85.4 | 66 | 50.5 | 47.6 | 51.5 | 68 | 95.1 | 49.5 | 66 | 52 |
| 25–30 yrs (n = 94) | 93.6 | 87.2 | 77.7 | 76.6 | 80.9 | 85.1 | 97.8 | 63.8 | 76.6 | 61.3 |
| 31–43 yrs (n = 102) | 97.1 | 87.1 | 69.6 | 72.5 | 69.6 | 86.3 | 99 | 75.2 | 73.3 | 61.1 |
| $\chi 2$ (df = 2) | 9.88 | 18.78 | 17.2 | 21.92 | 19.69 | 13.15 | 3.11 | 14.52 | 2.88 | 2.3 |
| p-value | **0.007** | **0.001** | **0.001** | **0.001** | **0.001** | **0.001** | 0.21 | **0.001** | 0.24 | 0.32 |
| **Education Level (Total)** | 92 | 79.9 | 65.6 | 65.2 | 66.9 | 79.6 | 97.3 | 62.8 | 71.8 | 57.9 |
| Did not complete HS (n = 151) | 94.7 | 86 | 65.3 | 71.3 | 66.7 | 83.3 | 98 | 65.8 | 67.3 | 59.2 |
| HS diploma/cert (n = 101) | 90.1 | 69 | 64.4 | 59.4 | 65.3 | 77.2 | 97 | 59.4 | 76.2 | 56.4 |
| Some college/ beyond (n = 48) | 87.5 | 83.3 | 68.8 | 58.3 | 70.8 | 72.9 | 95.7 | 60.4 | 76.6 | 57.4 |
| $\chi 2$ (df = 2) | 3.26 | 11.21 | 0.29 | 4.98 | 0.45 | 2.96 | 0.73 | 1.18 | 3 | 0.18 |
| p-value | 0.2 | **0.004** | 0.87 | 0.08 | 0.8 | 0.23 | 0.69 | 0.55 | 0.22 | 0.91 |
| **Marital status (Total)** | 91.9 | 79.7 | 65.3 | 65 | 66.7 | 79.5 | 97.3 | 63.2 | 71.6 | 57.6 |
| Single (n = 86) | 89.4 | 76.5 | 57.6 | 57.6 | 55.3 | 69.4 | 97.6 | 54.8 | 61.2 | 51.8 |
| Couple status (n = 212) | 92.9 | 81 | 68.4 | 67.9 | 71.2 | 83.5 | 97.1 | 66.5 | 75.8 | 60 |
| $\chi 2$ (df = 1) | 1.01 | 0.78 | 3.1 | 2.82 | 6.93 | 7.37 | 0.61 | 3.57 | 6.4 | 1.62 |
| p-value | 0.32 | 0.38 | 0.08 | 0.63 | **0.008** | **0.007** | 0.8 | 0.059 | **0.01** | 0.2 |

*Note*: Income and pregnancy (unintentional vs. intentional) were not associated with any of the statements, hence not shown in the table.

[*] Questions on Knowledge of Zika Virus.

a. Are you aware that Zika virus is transmitted primarily by Aedes aegypti mosquito bites?

b. Mosquitoes that spread Zika virus bite mostly during the daytime but can also bite at night.

c. Are you aware of the countries where Zika is currently and rapidly spreading?

d. Are you aware of what symptoms to look for if someone has Zika virus infection including fever, rash and pink eye?

e. Are you aware that there is no specific antiviral treatment or vaccine available for Zika virus disease?

f. Are you aware that if a pregnant woman contracts Zika virus babies are born a condition called "microcephaly", where babies are born with small heads and underdeveloped brain?

g. Do you know that the best way to prevent Zika virus infection is to avoid mosquito bites?

h. Do you know that a man or a woman with Zika virus can pass it to his sex partners during vaginal, anal or oral (mouth-to-penis) sex without a condom?

i. Did you hear about Zika virus scare before you got pregnant?

j. Did you and or your husband/partner hear or knew about Zika virus but did not worry/care to postpone getting pregnant?

condom, to delay at least 8 weeks, and 6 months if the male partner has symptomatic ZIKV infection, avoid sex for the duration of their pregnancy, and protective measures against mosquito bites (e.g., wearing long-sleeved shirts/long pants; staying indoors with protected screens; use of a mosquito bed net, and insect repellents) to prevent mosquito bites both during travel and for 3 weeks after returning to their home country. Maternal demographic factors (e.g., age, country of birth, education, income, marital status) and clinic type (FQHC versus Private) are independent correlates of interest in this study.

### Data analysis

Data were entered and cleaned for missing or invalid values in SPSS (v25). Univariate analyses included frequency distribution, proportions, mean, standard deviation, and percentages of study variables. Based on the distribution, categories in a given variable were regrouped to obtain meaningful counts for robust estimates. Cross-tabulations with the option of the two-tailed chi-square test were used (p values < .05) to identify significant bivariate associations between dependent and demographic factors (e.g., age, education, couple status, income levels). Multiple logistic regression analyses were conducted to examine factors associated with the dependent variables (each of the ten statements). Adjusted odds ratios (AOR) were calculated, and the level of significance for testing of each model was set to an alpha of .05, and 95% confidence intervals (CIs) are reported. However, we set a higher significance threshold (alpha .05/5 independent predictors in the model = adjusted alpha = .01) for individual comparisons to compensate for the number of inferences being made. Post-hoc power analysis of the study yielded 0.92 power computed based on two independent groups comparison with dichotomous response with alpha at 0.05.

## Results

### Participant demographics

The mean age of the mothers (N = 300) was 27.3 years (SD ± 6.03); with the range 18 to 43. Nearly two-thirds (62%) of the mothers were foreign-born, of which the majority were born in Mexico (80%) and remaining in other Latin American and Spanish-speaking countries (e.g., El Salvador, Guatemala, Honduras, Bolivia, Columbia, Cuba). Nearly one-half (47%) of the households spoke only Spanish at home, 28.7% spoke only English, and 24.3% spoke both languages. Fifty percent of the mothers did not complete high school, two-thirds (65.3%) are stay at home mothers. The majority (71%) were living with their spouse or a partner (couple status) and 29% were single (unmarried/divorced/ widowed) mothers. For more than one half (57%) of households, the monthly income was <$1,500 a month.

### Prenatal care and information sources

One in four mothers reported that the current pregnancy was their first child. Fifty-five percent of the mothers planned for current pregnancy and wanted the child, 32.4% mistimed or wanted to wait, and 12.3% reported it as unwanted pregnancy. At the time of the survey, one-fourth (23%) of the women were in the first trimester, and over one half (61%) were in the second trimester of their pregnancy. In terms of seeking prenatal care, one-half of the mothers reported visiting their physician by eight weeks into their pregnancy. The majority of women (92.5%) did not miss any of their prenatal visits.

One half (49%) of the women preferred receiving health-related information in Spanish, 39% in English, and the remaining (12%) indicated no preference. For almost all mothers (99%), nurse/physician is the primary source of information for health, nutrition, pregnancy, and the birthing process followed by their families (34%) and the media or internet (23%). Only a small percentage of the mothers (20%) also sought information from other sources (e.g. clinics, books/magazines, and friends).

### CDC recommendations concerning ZIKV transmission

The study results show that the mothers are aware of CDC specific recommendations concerning ZIKV. The majority of the mothers were aware of the travel ban for pregnant women to the ZIKV threat areas (77%); to avoid sexual transmission (76%); couples who want to get

pregnant should wait at least 8 weeks if traveled to the ZIKV outbreak area (e.g. Brazil) and to wait 6 months if the male partner has symptomatic ZIKV infection (76%); men should use condoms or avoid sex for the duration of their partner's pregnancy (81%); travelers should take protective measures (e.g. wearing long-sleeved shirts/long pants, staying indoors with protected screens, use of a mosquito bed net, and insect repellents) to prevent mosquito bites both during travel and for 3 weeks after returning to their home country (89%).

## ZKIV knowledge

ZIKV knowledge and behavior were assessed using ten statements that were formulated based on the information provided by the CDC. Of the women surveyed (Fig 1), 97% knew how to protect themselves from mosquito bites. The majority of women were aware that the virus is transmitted by *Aedes Aegypti* mosquito (both day and night) and via sexual intercourse. Those surveyed were also aware of the condition microcephaly and that there is no vaccination for the virus. More than one half (58%) reported that they knew or heard about the ZIKV outbreak. More than one half (55.3%) of the mothers reported that they wanted and intentionally got pregnant (n = 162). However, 61% (n = 95) of these women heard or know about ZIKV and did not worry or postponed the pregnancy.

We also created a composite score by summing the positive responses to eight knowledge questions (items a through h). Nearly two-thirds (61%) reported having knowledge in six or more items, 28% between 3–5 items and very small percentage (5%) reported having knowledge in two or less than two items.

## Correlates of ZIKV knowledge

Independent correlates were identified by using bivariate (using chi-square test) and multiple logistic regression analyses. A positive response to each statement was coded as 1 (yes) indicating the mother's knowledge specified in the statement. In general, results from bivariate analyses show significant associations between the country of birth, age, marital status, and ZIKV knowledge (Table 1). A higher proportion of foreign-born mothers reported having knowledge in eight out of ten knowledge areas compared with the mothers born in the United States. Compared to older women, significantly lower percentage of younger mothers (18–24 age group) reported knowledge in many of the areas. Women who live with their spouse or partner (couple status) seemed to be aware of microcephaly, transmission between sexual partners, and aware that there is no treatment for the ZIKV infection. Income and intentionality of pregnancy (unintentional vs. intentional) were not associated with any of the knowledge areas.

For multiple logistic regression, each dependent variable ("yes" response to each statement) was regressed on country of birth, age, education, income, and couple status (yes/no). Foreign-born women are between 2.5 to 3.0 times more likely to have knowledge concerning ZIKV, e.g., aware that mosquitoes can bite even at night, aware of symptoms, aware that there is no treatment, and aware of microcephaly condition in babies. However, younger mothers (18–24) are less likely to have knowledge concerning the ZIKV symptoms (fever, rash and pink eye) and transmission of infection via unprotected sexual behavior either vaginal, anal or oral (mouth-to-penis), compared to older women (Table 2).

## Clinic-based differences in ZIKV information dissemination

Overall, less than one-half of the mothers (48%) reported that HCPs specifically educated them about ZIKV and its effects on the baby. More than one third (39%) of the mothers neither received nor asked for information. Regarding travel history or plans to travel where the ZIKV was active, 47% of the mothers said their physician or nurse did not ask about travel

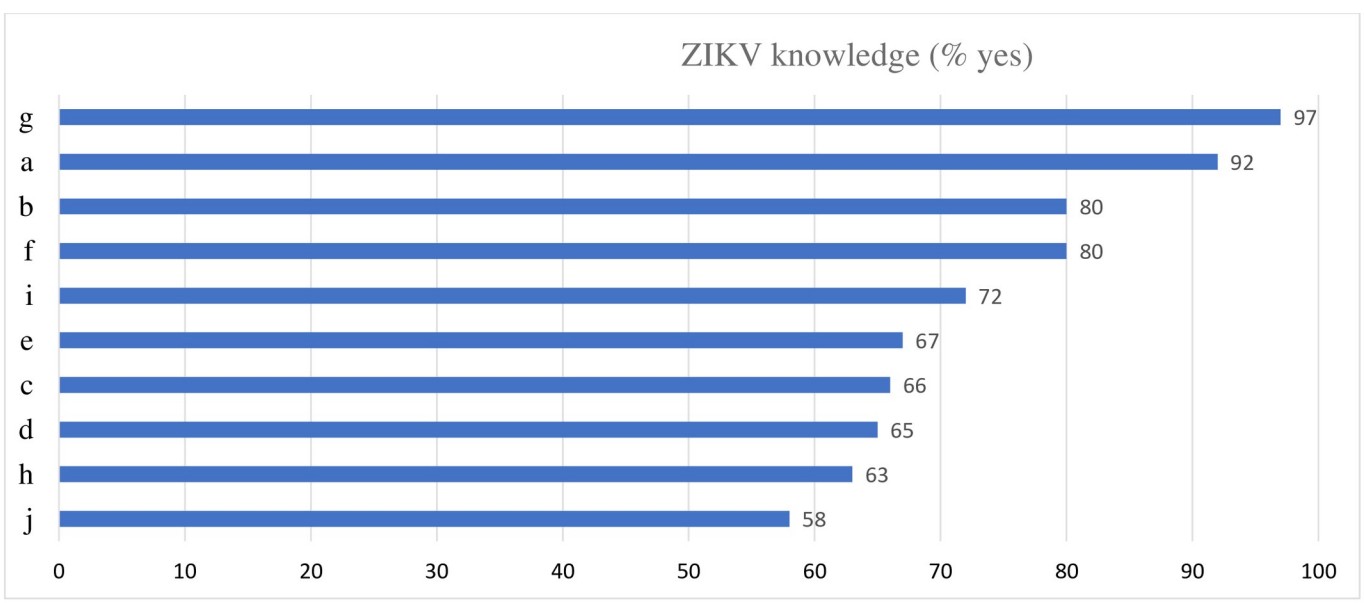

ZIKV knowledge (% yes)

- a. Are you aware that Zika virus is transmitted primarily by Aedes aegypti mosquito bites?
- b. Mosquitoes that spread Zika virus bite mostly during the daytime but can also bite at night.
- c. Are you aware of the countries where Zika is currently and rapidly spreading?
- d. Are you aware of what symptoms to look for if someone has Zika virus infection including fever, rash and pink eye?
- e. Are you aware that there is no specific antiviral treatment or vaccine available for Zika virus disease?
- f. Are you aware that if a pregnant woman contracts Zika virus babies are born a condition called "microcephaly", where babies are born with small heads and underdeveloped brain?
- g. Do you know that the best way to prevent Zika virus infection is to avoid mosquito bites?
- h. Do you know that a man or a woman with Zika virus can pass it to his sex partners during vaginal, anal or oral (mouth-to-penis) sex without a condom?
- i. Did you hear about Zika virus scare before you got pregnant?
- j. Did you and or your husband/partner hear or knew about Zika virus but did not worry/care to postpone getting pregnant?

**Fig 1. Zika Virus Knowledge among pregnant Latinas (% of yes).**

plans or they volunteered their travel plans. Clinic-based differences were observed in the proportion of mothers who were warned about ZIKV, an inquiry into mothers' travel history or plans, and the language in which health and ZIKV information were shared or displayed. Results indicate that the community health center (FQHC) neither actively inquired about mothers' travel plans ($\chi 2$ = 12.88; df = 2; p = .002) to places where ZIKV was active nor warned about travelling to places where ZIKV virus was active ($\chi 2$ = 5.45; df = 1; p = .02). Overall, less than half (45%) of the mothers reported that their physician/nurse specifically talked to them about safe sexual practices (e.g. use of a condom, abstinence). Forty-eight percent of women did indicate that they practice safe sex.

Significant clinic-based differences were also found in the language in which ZIKV information was displayed ($\chi 2$ = 29.18; df = 3; p = .001) and the language in which mothers personally received ZIKV information ($\chi 2$ = 35.10; df = 2; p = .001). A higher proportion of women who visited FQHC (59.5%), reported that the clinic displayed printed information (posters or handouts) and also received information personally in both languages (21%) or in Spanish (51%). A higher percentage of women from private clinics (29%) reported not seeing ZIKV

**Table 2. Results from multiple logistic regression -Adjusted Odds Ratios (AOR) and 95% Confidence Intervals (CI).**

| Correlates | a | b | c | d | e | f | g | h | i * | j * |
|---|---|---|---|---|---|---|---|---|---|---|
| | | | | | **Knowledge Statements** | | | | | |
| **Country of birth** | | | | | | | | | | |
| (ref: USA) | | | | | | | | | | |
| Foreign-born | 2.61 | 2.46 | 1.5 | 2.7 | 2.4 | 2.96 | 4.34 | 1.31 | 1.25 | 0.63 |
| 95% CI | (1.0–6.9) | (1.28,4.70) | (0.86,2.61) | (1.54–4.67) | (1.33–4.02) | (1.54–5.68) | (0.75,25.1) | (0.76,2.24) | (0.37,4.17) | (0.89,1.67) |
| p-value | 0.05 | **0.007** | 0.16 | **0.001** | **0.003** | **0.001** | 0.10 | 0.34 | 0.63 | 0.63 |
| **Age (years)** | | | | | | | | | | |
| (ref: 31+) | | | | | | | | | | |
| 18–24 | 0.25 | 0.41 | 0.47 | 0.45 | 0.58 | 0.45 | 0.30 | 0.36 | 0.81 | 0.48 |
| 95% CI | (0.07,0.95) | (0.19–0.88) | (0.25–0.88) | (0.24–0.85) | (0.31,1.01) | (0.21–0.97) | (.03,2.94) | (0.19,0.68) | (0.23,2.88) | (0.15,1.53) |
| p-value | 0.04 | 0.23 | 0.02 | **0.01** | 0.09 | 0.04 | 0.30 | **0.002** | 0.68 | 0.10 |
| 25–30 | 0.43 | 1.14 | 1.47 | 1.31 | 1.88 | 0.88 | 0.50 | 0.6 | 1.05 | 1.37 |
| 95% CI | (0.10–1.82) | (0.48,2.73) | (0.76,2.82) | (0.67,2.57) | (0.94,3.75) | (0.38,2.03) | (.04,5.85) | (0.32,1.13) | (0.32,3.48) | (0.45,4.13) |
| p-value | 0.25 | 0.76 | 0.25 | 0.43 | 0.07 | 0.77 | 0.58 | 0.11 | 0.92 | 0.47 |
| **Couple Status** | | | | | | | | | | |
| (ref: Couple) | | | | | | | | | | |
| Single | 1.1 | 1.07 | 0.78 | 0.9 | 0.63 | 0.56 | 2.24 | 0.66 | 0.78 | 1.70 |
| 95% CI | (0.42,2.87) | (0.54,2.12) | (0.44,1.37) | (0.51,1.61) | (0.36,1.13) | (0.29,1.08) | (0.39,12.7) | (0.38,1.15) | (0.23,2.63) | (0.53,5.47) |
| p-value | 0.85 | 0.84 | 0.38 | 0.73 | 0.12 | 0.08 | 0.36 | 0.14 | 0.59 | 0.24 |

Note:

§Adjusted for education and income.

* For items I and j, only those who reported planned pregnancy was considered for analysis.

Adjusted Alpha level .01is considered for significance.

information displayed anywhere compared to women from FQHC. Mothers also wanted more information on ZIKV including testing for the virus, protection and/or prevention, and information on vaccination against the virus (Table 3).

## ZIKV information sources

Mothers in the study reported on various sources of information through which they learned about ZIKV (Fig 2). Unlike primary source of pregnancy and health related information, which is often a physician or nurse, most mothers sought ZIKV information via the media/ internet (65%), one-third from a physician or nurse (23%) followed by clinic and family (20%), books/magazines (16%) or friends (13%).

## Travel history, plans, and concerns of the mothers

Only a few women (n = 12) traveled to countries where ZIKV was active including Mexico (n = 9) during the three months prior to their pregnancy. Since becoming pregnant, ten women traveled to other countries: Aruba, Canada, Mexico, and Puerto Rico. Traveling with in the United States was also sparse: 11% traveled three months prior to pregnancy and only 7% (n = 27) traveled since they became pregnant. Spouses of ten of the mothers also traveled to other countries (e.g., Mexico, Puerto Rico, Aruba) three months prior to pregnancy and nine spouses since their partner became pregnant. A considerable percentage of the mothers were "very afraid" (44%) or "somewhat afraid" (38%) to travel to places where there was a high risk of contracting the virus, and the rest were not at all afraid (18%). Ten percent said they

**Table 3. Clinic based differences in educating and sharing Zika related information.**

| | Total | FQHC | Private | χ2 (df) | p-value |
|---|---|---|---|---|---|
| | (N = 300) | (n = 192) | (n = 108) | | |
| | % | % | % | | |
| 1. Staff specifically educated women about Zika virus and its effect on the baby | | | | | |
| a. Intentionally educated without asking for it | 48 | 49.2 | 45.1 | 0.66 (2) | 0.72 |
| b. Educated about Zika virus only after I asked for the information | 13 | 11.9 | 14.7 | | |
| c. They did not give you any information and I did not ask | 39 | 38.9 | 40.2 | | |
| 2. Physician/nurse specifically asked about travel history or travel plans to a place where Zika virus outbreak is active | | | | | |
| a. I told them only when they asked me about it. | 43.8 | 36 | 57.8 | 12.88 (2) | **0.002** |
| b. I told them even if they did not ask me. | 9.4 | 10.2 | 7.8 | | |
| c. They did not ask, and I did not tell | 46.9 | 53.8 | 34.3 | | |
| 3. Warned about travelling to places with Zika outbreak (yes %) | 48.5 | 43.4 | 57.5 | 5.45 (1) | **0.02** |
| 4. Physician/nurse specifically talked about safe sexual practices (e.g., using a condom consistently or avoiding sex) during pregnancy if you think you or your partner are exposed to Zika virus? (yes %) | 45.4 | 46.6 | 43.3 | 0.30 (1) | 0.583 |
| 5. Information handed over personally | | | | | |
| a. Spanish | 42.2 | 50.6 | 26.9 | 35.1 (2) | **0.001** |
| b. English | 41.4 | 28.2 | 65.6 | | |
| c. Both Spanish and English | 16.3 | 21.2 | 7.5 | | |
| 6. Displayed posters, brochures or handouts (Zika related information) | | | | | |
| a. Spanish | 13.5 | 16.3 | 8.2 | 29.18 (3) | **0.001** |
| b. English | 16.7 | 11.1 | 27.6 | | |
| c. Both Spanish and English | 51.4 | 59.5 | 35.7 | | |
| d. Did not display any information on Zika | 18.4 | 13.2 | 28.6 | | |

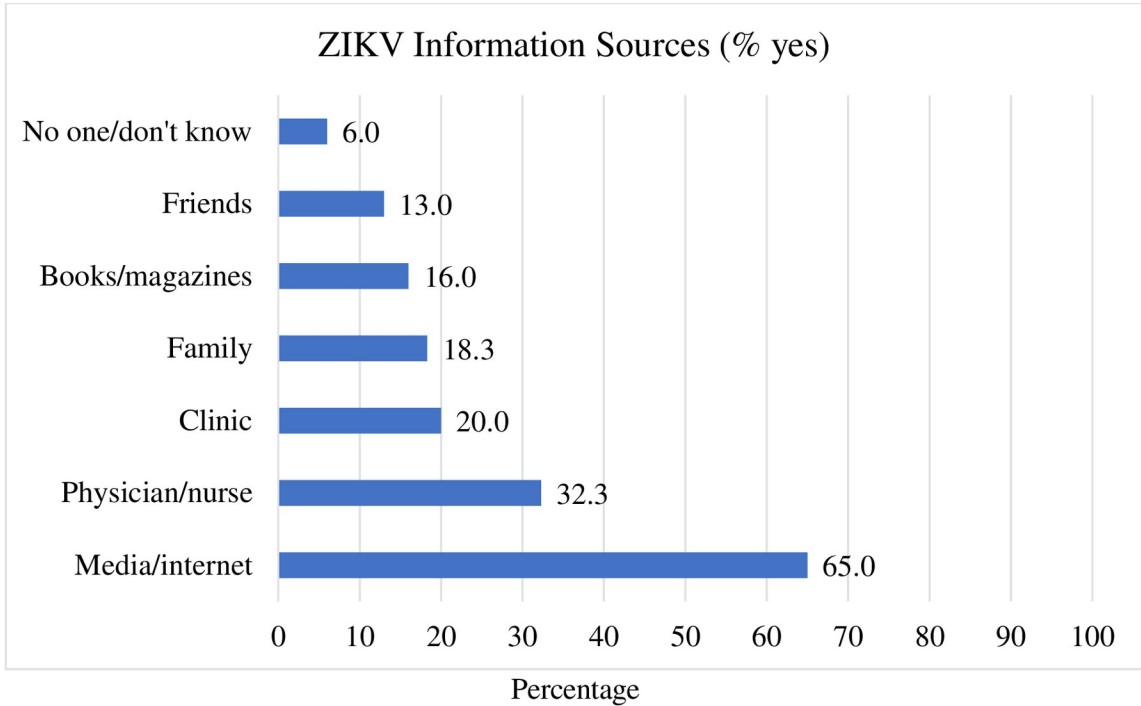

**Fig 2. Information sources for Zika Virus (N = 300).**

would not change any travel plans to destinations where there was a ZIKV outbreak. Four women indicated that they had plans to visit Mexico before the baby was born.

## Discussion

This study investigated knowledge and awareness of the ZIKV among underserved pregnant Latinas. The study addressed two research questions, whether demographic factors have an influence on ZIKV knowledge and whether there is a difference in provider communication and information dissemination based on the clinic type (FQHC versus Private). In general, our study revealed that almost all mothers have a basic understanding of the ZIKV transmission, ZIKV prevention strategies as well as were able to identify countries where ZIKV was active at the time of the survey.

The study points to some important demographic factors that may influence mothers' knowledge concerning ZIKV. A significantly lower proportion of women in the youngest age group (18–24) reported having knowledge on many of the statements. Foreign-born women (born in outbreak areas) were nearly three times more likely to have knowledge of ZIKV transmission via mosquitoes, symptoms, and awareness of the microcephaly condition in babies. The most recent study by Berenson et al. [20] suggests that this difference could be due to counseling and discussions with their family members still living in their birth countries. However, in our study, incorrect information, e.g. using sunblock or receiving vaccination against ZIKV was also mentioned by mothers. HCPs must address if incorrect information is being spread and point to the local health authorities for clarification. Factual knowledge must be imparted through community level education via social service organizations [21], particularly those serving low-income vulnerable populations. HCPs have the opportunity to collaborate with local epidemiologists and health educators to address such problems and stimulate change.

Women with partners seemed to be aware of the microcephaly condition in babies, transmission between sexual partners, and the lack of treatment for ZKV. Although single parent status indicated lower levels of knowledge in some areas at bivariate level, multivariate analyses did not suggest knowledge dependence on couple status. A recent study reported that age, marital status, education, and native language were unrelated to knowledge, but noted gender disparity in ZIKV knowledge. Women were significantly more likely than men to know about sexual transmission and maternal transmission to the fetus. Greater awareness of the risk of sexual transmission of ZIKV is needed, especially for men [22].

Two-thirds of the mothers were foreign-born, and it is reasonable to expect frequent travel to their home country. A small percentage of mothers did report travel to Latin American countries three months prior to their pregnancy. Frequent travel during the outbreak seasons will increase the risk of ZIKV exposure. Based on the geographic exposure data, Rao et al. reported that Mexico (44%), the Caribbean (17%), North America (16%), South America (13%), and Central America (9%) were the most common areas in which potential exposure occurred in their study sample [23]. Mistimed or unwanted pregnancies may create further complications as there is a possibility to travel to these high-risk areas and may not recognize the importance of protecting themselves from mosquito bites or practicing safe sex. Nearly two-thirds (61%) of the mothers who planned their pregnancy knew about the outbreak, but may not have perceived the risk of contracting the virus as they live in the northern part of Texas. In our sample, few mothers and their partners reported travel plans, which may explain why the women may not perceive the risk even in the case of a planned pregnancy.

Nearly one-half of the households spoke only Spanish (47%), 50% of the mothers did not complete high school and two-thirds (65.3%) are stay at home mothers. Information regarding

ZIKV evolved rapidly during the 2015 outbreak and the public health authorities used media/internet as the major means of dissemination. Latinas from low-socioeconomic backgrounds may have language difficulty as most information is available in English. Further, much of the clinical information on ZIKV is posted on health agencies' websites (World Health Organization/CDC/Local Health Departments), where Latino families with limited resources may not be able to access. Studies show racial/ethnic disparities in overall internet use and its use to access health information. Overall, internet use is high among non-Hispanic white (92.7%) and non-Hispanic African American (92.9%) women in comparison to Hispanic women (67.5%) [24, 25]. Additionally, higher levels of internet use were reported among mothers who are comfortable speaking English compared to those who spoke only Spanish. These groups are viewed as underrepresented populations and difficult to reach or involve in research or public health programmes [26], not only because of language barrier and lower trust in media, but also because they report relatively low use of various media channels. These findings have important implications for communicating health concerns toward non-native speakers of English and Hispanics in particular [27]. A recent report by Diaz [28], suggested that Latino tended to discuss the Zika virus in Spanish (89%), versus English (10%) and bilingually (1%). Thus, delivering health messages and appropriate counseling in native languages is the culturally relevant way for positive behavior change and outcomes.

Clinical staff (nurses and physicians) and family members are the main sources for pregnancy and ZIKV related information. The majority of the study participants (65%) sought information via the media/internet and one third (33%) sought information from a physician/nurse or the clinic. Nurses and clinicians, especially those who work with women of childbearing age, should take a pivotal role in disseminating accurate information and be proactive in identifying potential risk. Healthcare professionals believe that it is crucial for patients to be educated about health for optimal outcomes. To improve knowledge in the targeted group, HCPs need to know about the sources from where their patients seek information [29, 30]. Although the internet may be an easily available source of health information, disparities in accessibility may exist. The internet could serve as a supplemental resource to receiving health information directly from care providers for making informed decisions [31]. This may require HCPs to be knowledgeable about internet resources including official websites of health authorities (e.g. CDC and World Health Organization) where relevant and recent information is available.

FQHC clinics play an important role in providing care to the most vulnerable groups and fill an expansive void in America's health care system. Clinic-based differences suggest that FQHCs must develop a standard protocol (a) to inquire about travel history or plans; (b) to provide warnings concerning ZIKV outbreak areas, and (c) to disseminate health and ZIKV information in various languages spoken among their patients. Overall, only less than half of the mothers reported that physician/nurse specifically talked to them about safe sexual practices (use of a condom, abstinence, etc.) and inquired about the mothers' current sexual practices. Providers should identify patients likely to become pregnant and travel to high-risk areas, inquire about partner travel history, and offer culturally appropriate ZIKV risk counseling [10]. Conversations around sexual behavior is a delicate matter and must be discussed in a culturally sensitive manner. For example, the CDC guidelines regarding the risk of contracting ZIKV via the sexual transmission and conversations around safe sexual practices to prevent pregnancy must be discussed with the couple together, not with only one partner. For persons with possible ZIKV exposure who are planning to conceive and interim guidance to prevent transmission through sexual contact must be dealt with utmost cultural sensitivity. Couples with possible virus exposure, who are not pregnant and do not plan to become pregnant, should be encouraged to minimize their risk for sexual transmission by practicing safe sex.

FQHCs are the first line of community-based health care services in under-served areas. In addition to providing web-based ZIKV related information on their official websites, CDC and HRSA should support and supervise community-based clinics to make sure that the clinical information is conveyed to the targeted vulnerable at-risk populations.

## Strengths and limitations

Our study has three important and noteworthy features. First, our study sample is from low income, underserved, undocumented Latinas that are population and are seldom researched. Second, the study takes advantage of a large sample that is inclusive of the majority (85%) of the low-income pregnant Latinas in the county, especially, at the county's FQHC. To account for language vulnerability, the survey was available both in English and Spanish. Lastly, only a small number of mothers (n = 5) refused because of lack of time. We also provided compensation for their time to attain a higher response rate.

The study suffers from weaknesses that are typical of survey research. As data were collected cross-sectionally, our understanding of causal relationships among the study variables is limited. The non-probability sampling of pregnant Latinas (predominately of Mexican descent) from a single geographic region limits the generalizability of study findings beyond the North Texas region. The study location is not a typical border community where greater than 50% of the population is Latino. There is a greater variation in percentage of the Hispanic population across Texas counties with much higher proportions, between 70 and 90%, in the counties bordering Mexico. Further, the number of women that do not seek prenatal care in this vulnerable population is unknown; hence the study findings cannot be extended to all women in this targeted population.

Our findings could also have been affected by subjective interpretation, access to information, and recall bias. Since recruitment was conducted in the clinics, there could have been a change in the type of information provided once the physician/nursing staff were aware that the study was being conducted. Such behavior change could inflate positive responses. The dependent variable, knowledge in the form of a statements used binary responses (with yes/no) that might have resulted in potential response bias resulting from over reporting. Findings from our study have limited generalizability to Latinas in other regions of the country as they might belong to other Hispanic subgroups (e.g. Cuban, Puerto Rican), self-reporting bias, and a lack of survey validation as an indicator of English language proficiency.

## Study implications

The study findings have great potential in developing appropriate protocols and strategies to ensure better and timely communication between the clinic staff and mothers during health emergencies. Due to time concerns, HCPs should enhance the rapid delivery of crucial information, information of credible websites, and be proactive in providing patient education via brochures, personal chat sessions, webinars, and videos. The study findings suggest that FQHCs must develop a standardized protocol to provide information, warnings verbally and visually (e.g. audiovisual, infographics) during health emergencies in multiple languages that cater to their targeted populations. In addition to providing web-based ZIKV related information, local health departments (LHDs), CDC, and HRSA should play an active role in supervising community-based clinics to ensure that the clinical information is conveyed to the at-risk populations.

Cultural factors, including immigration or place of birth, are essential in understanding the use of health information sources. In particular, the concern of outbreaks such as the ZIKV and their decision-making about pregnancy and child health. Research supports the notion

that ZIKV could be transmitted sexually [32, 33]. Reproductive rights and pregnancy decision-making are the topics that are closely tied to religious and other cultural factors that limit access to contraception, which was highly recommended during the ZIKV outbreak [34, 35]. Research suggests that knowledge of the sexual transmissibility of ZIKV significantly increases the odds of taking a preventive action against the infection, especially condom use or sexual abstention [36]. Borges et al. in their Brazilian study reported that there were missed opportunities for prevention of perinatal transmission of the virus through behavioral change, including effective contraception to prevent pregnancy and condoms to prevent perinatal transmission, as a complement to vector control [37]. Existing decision-making patterns lead to an increased risk of ZIKV exposure. Long-term response planning must include educating men and young people concerning ZIKV, in addition to engaging with broader societal challenges to gender inequity [38]. Prenatal clinics are ideal settings for Latina mothers to seek much needed, credible health information. The Texas Zika Virus Preparedness and Response Plan [39] urges for clear and timely messaging between medical and public health professionals and monitors that the messages reach intended vulnerable populations (e.g., underserved populations along the Texas-Mexico border, pregnant women, and travelers). Public campaigns to enhance awareness and coordinating media messaging with LHDs to promote consistency are strongly urged.

Our research focused on the need for health care providers delivering timely information about ZIKV to the underserved families that visit community healthcare clinics. With the current existing COVID-19 health emergency across the globe, pregnancy planning and prevention becomes more pertinent. Medical research to study the impact of COVID-19 on pregnancy is still unfolding. Research on maternal health, possible vertical transmission, in utero exposure, pregnancy outcomes, long-term health effects on infants is still evolving [40, 41]. Our study findings will lay the foundation for a larger study for the development of appropriate protocols, strategies, and health campaigns to ensure timely communication, and alternate modalities of patient education and risk communication that can supplement in-person conversations [10]. Healthcare professionals must consider utilizing the best mode of communication for the community they serve. Just as our research found with ZIKV, dissemination of information on COVID-19 is crucial. Lack of information, timely access, and credible information is key in any public health crisis.

## Conclusions

Our results indicate an overall decent understanding of ZIKV symptoms, possible complications, transmission modes, and current recommended prevention guidelines. Younger women and those born in the United States are negatively associated with knowledge of ZIKV. There is a need for implementing future public health interventions that focus on protection against ZIKV transmission, particularly in community health centers serving underserved populations. Enhancing ZIKV awareness, in particular, among women of reproductive age, is crucial for the reduction of babies born with microcephaly and associated congenital abnormalities. Our study also underscores the need for HCPs, in particular, the nursing and frontline workers, to be trained and prepared to educate families that are at higher risk of exposure. Although HCPs may lack sufficient time to discuss concerns with every patient, they may consider providing patient education in other ways. Providers should consider taking advantage of using alternate modalities including educational brochures, active, enhanced information dissemination via podcasts on providers' websites, emails, and text messaging to supplement in-person exchanges, particularly during a public health emergency.

## Acknowledgments

We are thankful to the prenatal clinics, staff who helped to collect data, and all the mothers who took part in the study.

## Author Contributions

**Conceptualization:** Suhasini Ramisetty-Mikler.

**Data curation:** LeAnn Boyce.

**Formal analysis:** Suhasini Ramisetty-Mikler, LeAnn Boyce.

**Funding acquisition:** Suhasini Ramisetty-Mikler.

**Investigation:** Suhasini Ramisetty-Mikler.

**Methodology:** Suhasini Ramisetty-Mikler.

**Project administration:** Suhasini Ramisetty-Mikler, LeAnn Boyce.

**Supervision:** Suhasini Ramisetty-Mikler.

**Writing – original draft:** Suhasini Ramisetty-Mikler.

**Writing – review & editing:** Suhasini Ramisetty-Mikler, LeAnn Boyce.

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
