## [Decision Letter · Decision Letter 0]

3 Apr 2020

PONE-D-19-24828

Communicating the Risk of Contracting Zika Virus to Low Income Underserved Pregnant Latinas: A Clinic-based Study

PLOS ONE

Dear Dr. RAMISETTY-MIKLER,

Thank you for submitting your manuscript to PLOS ONE. After careful consideration, we feel that it has merit but does not fully meet PLOS ONE’s publication criteria as it currently stands. Therefore, we invite you to submit a revised version of the manuscript that addresses the points raised during the review process.

Until now, the decision is based on the recommendations raised by one reviewer (see below). You may consider to incorporate these changes so that we can hand the revised version to the next round of the review process, where we aim to have at least two reviewer reports.

We would appreciate receiving your revised manuscript by May 18 2020 11:59PM. To enhance the reproducibility of your results, we recommend that if applicable you deposit your laboratory protocols in protocols.io, where a protocol can be assigned its own identifier (DOI) such that it can be cited independently in the future. For instructions see: http://journals.plos.org/plosone/s/submission-guidelines#loc-laboratory-protocols

We look forward to receiving your revised manuscript.

Kind regards,

Florian Fischer

Academic Editor

PLOS ONE

2. Please include additional information regarding the survey or questionnaire used in the study and ensure that you have provided sufficient details that others could replicate the analyses. For instance, if you developed a questionnaire as part of this study and it is not under a copyright more restrictive than CC-BY, please include a copy, in both the original language and English, as Supporting Information. Also, if this questionnaire was pre-tested please include details of these participants and upon how many the pre-testing occurred. Furthermore, please justify the sample size involved in this study and any information pertaining to this - e.g. was a power calculation performed prior to participant recruitment? Also, please refer to any post-hoc corrections following multiple comparisons that were performed, or explain why these were not included.

5. Your ethics statement must appear in the Methods section of your manuscript. If your ethics statement is written in any section besides the Methods, please move it to the Methods section and delete it from any other section. Please also ensure that your ethics statement is included in your manuscript, as the ethics section of your online submission will not be published alongside your manuscript.

6. Please upload a new copy of Figure 1-2 as the detail is not clear. Please follow the link for more information: http://blogs.PLOS.org/everyone/2011/05/10/how-to-check-your-manuscript-image-quality-in-editorial-manager/

Reviewers' comments:

Reviewer's Responses to Questions

**Comments to the Author**

1. Is the manuscript technically sound, and do the data support the conclusions?

Reviewer #1: Partly

2. Has the statistical analysis been performed appropriately and rigorously? 

Reviewer #1: Yes

3. Have the authors made all data underlying the findings in their manuscript fully available?

Reviewer #1: Yes

4. Is the manuscript presented in an intelligible fashion and written in standard English?

Reviewer #1: Yes

5. Review Comments to the Author

Reviewer #1: The paper entitled “Communicating the Risk of Contracting Zika Virus to Low Income Underserved Pregnant Latinas: A Clinic-based Study” addresses as important and often overlooked topic in Zika research, communication of risk to higher risk populations. Overall this paper is nicely written and provides insights into the clinician-patient interactions during and following the Zika pandemic. There are several things that could be done, however, that would improve the manuscript.

Introduction:

- It Is important to contextualize the research presented in the introduction with the dates and time-frames that they address. Knowledge and communication transformed throughout the pandemic and many of the articles cited appear to be from early on in the pandemic.

- Authors could update their introduction with findings from a broader geographic context; there are a number of other studies on Zika knowledge, attitudes and practices that could provide some useful background that were focused in Florida.

Methods:

- More details on study site are needed – north Texas county, distance from border, additional characteristics about the population in the country would be helpful; percent below poverty, percent born in other countries. The study predicates that there is a gap in knowledge in border communities yet the limited data on the area suggests this is not a typical border community where greater than 50% of the population are typically LatinX. Some context and discussion about the site will allow readers to understand the generalizability to the border communities.

- More description of the recruitment strategy and how they determined this clinic served 85% of Latina women in the county are warranted. How did they ensure women did not take the survey multiple times?

- What stage of pregnancy were the women in when they took the surveys? This could speak to risk – i.e. if it is October 2016 and you were pregnant in March 2016 that was before there were many documented local cases in the US.

- October 2016- Feb. 2018 is a long time period. It is important to include a variable that divides the data into time periods (no more than three given the smaller sample size) to examine how these data changed over time. My guess is Zika information dissemination feel rapidly throughout 2017. This time frame was very late in the pandemic and the timing should be discussed.

- Limitations of the dependent variable with yes/ no responses should be discussed including potential response bias. Individuals may have a tendency to over report their prior knowledge.

- While the single variable approach is great to determine the domains of knowledge that are weaker, an aggregate score could reveal patterns in overall general knowledge that could be further useful. Suggest conducting and including in supplementary data.

- Despite pilot testing the last knowledge question about delaying pregnancy is confusing. While interesting, it may be hard to interpret given it is a three part question – first asking about if they knew about Zika, then asking if it made them worry or care, and third asking if they delayed the pregnancy because of it. The limitations of this question should be discussed.

- The explanatory variables are quite limited and may be subject to confounding.

- Table 2 – items g and j should still be presented.

- Table 3- examining this by period of the study is interesting; Since recruitment was conducted in the clinics could their have been a change in the type of information provided given clinicians knew the study was ongoing?

Discussion

- The low rates of travel may explain why the women were not feeling at risk

- Single moms are only significantly lower on the planning and caring to postpone pregnancy – this is likely confounded by unintended pregnancy, if they were not planning to become pregnant then they would not be likely to postpone.

- Why did they not look at source of knowledge and knowledge associations, this would be more straightforward and address some of the questions about how women receive knowledge and their misperceptions

- Other limitations and biases as outlined above should be presented and discussed.

6. PLOS authors have the option to publish the peer review history of their article (what does this mean?). If published, this will include your full peer review and any attached files.

Reviewer #1: No

---

## [Author Response · Author response to Decision Letter 0]

27 Apr 2020

Response to Reviewers:

Communicating the Risk of Contracting Zika Virus to Low Income Underserved Pregnant Latinas: A Clinic-based Study

We received reviews from only one individual. The following are our responses (blue text) to reviewer's Questions. We used track changes option with pink color to mark edits and inserts in the text. 

 Reviewer #1: 

The paper entitled “Communicating the Risk of Contracting Zika Virus to Low Income Underserved Pregnant Latinas: A Clinic-based Study” addresses as important and often overlooked topic in Zika research, communication of risk to higher risk populations. Overall this paper is nicely written and provides insights into the clinician-patient interactions during and following the Zika pandemic. There are several things that could be done, however, that would improve the manuscript.

Introduction:

- It Is important to contextualize the research presented in the introduction with the dates and time-frames that they address. Knowledge and communication transformed throughout the pandemic and many of the articles cited appear to be from early on in the pandemic.

- Authors could update their introduction with findings from a broader geographic context; there are a number of other studies on Zika knowledge, attitudes and practices that could provide some useful background that were focused in Florida.

RESPONSE:

Added pertinent text as requested. We have included recent information concerning Zika knowledge, attitudes and practices under introduction, see pages 5 and 6.

Methods:

- More details on study site are needed – north Texas county, distance from border, additional characteristics about the population in the country would be helpful; percent below poverty, percent born in other countries. 

RESPONSE:

Added pertinent text as requested concerning the county and the state under participants section, see pages 7 and 8.

- More description of the recruitment strategy and how they determined this clinic served 85% of Latina women in the county are warranted. How did they ensure women did not take the survey multiple times?

RESPONSE:

Added pertinent text as under study participants, see page 7. 

- October 2016- Feb. 2018 is a long time period. It is important to include a variable that divides the data into time periods (no more than three given the smaller sample size) to examine how these data changed over time. My guess is Zika information dissemination feel rapidly throughout 2017. This time frame was very late in the pandemic and the timing should be discussed.

RESPONSE:

The study began in October 2016; however, the actual data collection was done between January 2017 and February2018. The first three months of the study were spent in locating the clinics and obtaining their approval, getting the study approvals from the university Institutional Review Board, and recruiting the mothers. We have included the text to clarify this timeline. See page 8.

- Justify the sample size involved in this study and any information pertaining to this - e.g. was a power calculation performed prior to participant recruitment? Also, please refer to any post-hoc corrections following multiple comparisons that were performed or explain why these were not included.

RESPONSE:

The sample size was determined based on the funding available. However, we have performed post-hoc power analysis and arrived study power at 0.92, which is high. We also considered significance level of adjusted alpha .01 (see page 9) for multiple comparisons. Adjusted odds ratios (AOR) were calculated, and level of significance for testing of each model was set to an alpha of .05, and 95% confidence intervals (CIs) are reported. However, we set a higher significance threshold (.05/5 independent predictors in the model= adjusted alpha = .01) for individual comparisons to compensate for the number of inferences being made, see page 10. 

- While the single variable approach is great to determine the domains of knowledge that are weaker, an aggregate score could reveal patterns in overall general knowledge that could be further useful. Suggest conducting and including in supplementary data.

RESPONSE:

We have created an aggregate score by summing the knowledge items in our earlier analyses. Since there was not much variation, (only less than 5% (n=14) reported having knowledge on two items or less. Hence, we did not report. As per the reviewer’s comment, we have now included the results now in the revised version. See page 12.

- The explanatory variables are quite limited and may be subject to confounding. 

RESPONSE:

Explanatory variables include individual and demographic factors that are typically collected in any survey (age, education, marital status, income, employment, and pregnancy-intentional/unintentional). We are not sure what other demographics the reviewer was referring to. It would have been nicer if he or she had mentioned them. We have used all the variables collected in the study.

- Table 2 – items g and j should still be presented.

RESPONSE:

We have now included results from the LR for items g and j in Table 2. See page 16.

Discussion

- The low rates of travel may explain why the women were not feeling at risk

- Single moms are only significantly lower on the planning and caring to postpone pregnancy – this is likely confounded by unintended pregnancy, if they were not planning to become pregnant then they would not be likely to postpone. 

RESPONSE:

We have now included pertinent information in the revised manuscript (page 21).

- Why did they not look at source of knowledge and knowledge associations, this would be more straightforward and address some of the questions about how women receive knowledge and their misperceptions

RESPONSE:

The response categories for the question that inquired sources is “check all that apply”, meaning the responses are not mutually exclusive. A person can obtain information via multiple sources simultaneously. For this reason, we cannot test associations between source of knowledge and knowledge.

LIMITATIONS

The study predicates that there is a gap in knowledge in border communities yet the limited data on the area suggests this is not a typical border community where greater than 50% of the population are typically LatinX. Some context and discussion about the site will allow readers to understand the generalizability to the border communities.

RESPONSE:

Added pertinent text as requested concerning the generalizability under limitations. See pages 23-24. 

- Limitations of the dependent variable with yes/ no responses should be discussed including potential response bias. Individuals may have a tendency to over report their prior knowledge.

RESPONSE:

We have now included this limitation in the text in page 24.

- Despite pilot testing the last knowledge question about delaying pregnancy is confusing. While interesting, it may be hard to interpret given it is a three part question – first asking about if they knew about Zika, then asking if it made them worry or care, and third asking if they delayed the pregnancy because of it. The limitations of this question should be discussed.

RESPONSE:

This question is not a three-part question and was not asked in sequence to prior questions. This is an independent question that asks for decision making based on perceived risk hence reflects behavior rather than knowledge. 

- Table 3-; Since recruitment was conducted in the clinics could their have been a change in the type of information provided given clinicians knew the study was ongoing?

RESPONSE:

We agree with the reviewer that there is a possibility of change in the behavior of clinicians over time. Clinicians neither directly involved in the study nor and knew about the specific items asked in the survey. Thus, we assume that they did not change their communication behavior. We have added this issue under limitations, see page 24.

---

## [Decision Letter · Decision Letter 1]

22 Sep 2020

PONE-D-19-24828R1

Communicating the Risk of Contracting Zika Virus to Low Income Underserved Pregnant Latinas: A Clinic-based Study

PLOS ONE

Dear Dr. RAMISETTY-MIKLER,

Thank you for submitting your manuscript to PLOS ONE. After careful consideration, we feel that it has merit but does not fully meet PLOS ONE’s publication criteria as it currently stands. Therefore, we invite you to submit a revised version of the manuscript that addresses the points raised during the review process.

We look forward to receiving your revised manuscript.

Kind regards,

Florian Fischer

Academic Editor

PLOS ONE

Reviewers' comments:

Reviewer's Responses to Questions

**Comments to the Author**

1. If the authors have adequately addressed your comments raised in a previous round of review and you feel that this manuscript is now acceptable for publication, you may indicate that here to bypass the “Comments to the Author” section, enter your conflict of interest statement in the “Confidential to Editor” section, and submit your "Accept" recommendation.

Reviewer #2: (No Response)

Reviewer #3: All comments have been addressed

2. Is the manuscript technically sound, and do the data support the conclusions?

Reviewer #2: Partly

Reviewer #3: Yes

3. Has the statistical analysis been performed appropriately and rigorously? 

Reviewer #2: Yes

Reviewer #3: Yes

4. Have the authors made all data underlying the findings in their manuscript fully available?

Reviewer #2: Yes

Reviewer #3: Yes

5. Is the manuscript presented in an intelligible fashion and written in standard English?

Reviewer #2: Yes

Reviewer #3: Yes

6. Review Comments to the Author

Reviewer #2: The study titled, “Communicating the Risk of Contracting Zika Virus to Low Income Underserved Pregnant Latinas: A Clinic-based Study” is a timely study on knowledge and health risk communications on the Zika outbreak with an underserved, high risk community. Several areas need further clarification. Below are detailed recommendations to enhance the manuscript.

Page 4

line 48

Replace “and” with “who”

line 49

State “Latin America” rather than “South America” since some foreign born participants were from Central America

Page 6

line 96

Reword to depict that there is a paucity of data on pregnant Latinas, state why they are an important population to study. Underserved pregnant Latinas are at higher risk for…(references)

Page 7

Lines 118-129

Authors do not provide inclusion and exclusion criteria for the study. For example, if literacy level was low, were the participants excluded from completing the survey? If participants as Brazilian or Spanish, were they excluded due to Spanish-language and/or European origin?

Page 8

lines 136-139

Please provide more details as to how potential participants were identified as “Latino.” Did they self-identify? Did the staff identify them based on their medical records? Did the staff ask them to self-identify for the purposes of this study? More detail is needed regarding how the ethnicity of the participants was determined for study inclusion. Provide a statement explaining the training of clinic staff to obtain informed consent.

Lines 151-159

Please provide more detail on how study participants completed surveys and how many questions the survey included. Were these completed with pencil and paper, link to online survey, etc. And if participants had a low literacy level, did they have a partner/friend/relative help them complete it or was it administered by trained study personnel in an interview format? More information is also needed on the survey responses and measures. Please describe how awareness, knowledge, travel behavior and information sources were measured in the survey. For example, were any of the questions open-ended, ranking, Likert scale items, etc.

Page 11

Lines 207-209

“Significantly higher proportion of 208 single mothers (57.8%) reported untimed/unwanted pregnancy compared to mothers who live with 209 their partner/spouse (38.9%) (2 = 8.58, df = 1, p=.003).”

This statement should be removed from the paper. It does not contribute to the author’s research questions, nor does it provide data for rich discussion. Further, it perpetuates racial/ethnic stereotypes.

Page 12

Lines 241-242

Remove qualifier “surprisingly.” The authors did not collect data on where geographically the conception occurred, therefore it cannot be assumed that they felt ZIKV was a risk that contributed to family planning.

Page 16

Table 2

”§Adjusted for education and income” – Education and Income often indicate the same measure (multicollinearity), health care access. Research question states to examine only “education” among the demographics listed. If multicollinearity did not exist, either add income into research question. Although, the authors have a stronger variable to that also measures health care access, clinic type. This variable is more pertinent in this analysis to establish differences (if any) between patients at both clinics.

Page 17

Lines 304-306

“Results indicate that the community health center (FQHC) neither actively inquired about mothers’ travel plans (2 =12.88; df=2; p = .002) nor warned about travelling to places where Zika virus was active (2 = 5.45; df=1; p = .02).”

In this statement, authors report that providers did not actively inquire about travel plans, however, please clarify that the question the participants were asked stated “Physician/nurse specifically asked about travel history or travel plans to a place where Zika virus outbreak is active.” Therefore, participants responded to the question of “travel to a place where Zika virus outbreak is active,” not just travel plans. Caution with interpretation of responses from this question. First, assumes that the patient knows where Zika virus is active when women in the younger age group possessed less knowledge on Zika in general. Second, the question is specific about “travel history or travel plans to a place where Zika virus outbreak is active,” not just travel plans in general.

Page 19

Lines 324-328 & Figure 2

It is unclear how the sources of information were assessed. Was this a select all that apply type of question or was it a ranking?

Lines 334-344

Same concern with this section. How were these questions measured or were they open ended? If so, is it possible that the participants had survey fatigue and skipped this section?

Page 20

Lines 353-355

“Although there was an awareness of the negative impact on the babies, some women did not worry or care to delay pregnancy, in particular among mothers who planned their pregnancy.”

Authors do not provide data supporting this statement in either the tables or figures, except for one statement regarding “Fifty-five percent of mothers planned for current pregnancy”. I suggest it be removed.

Line 364

Suggesting the HCPs “must” address misinformation is not reasonable. Health care providers each have roles, but they can collaborate with local health departments if misinformation is spreading through communities. Health departments are equipped with epidemiologists and health educators to address such problems, and HCPs have an opportunity to stimulate change.

Lines 364-366

“Factual knowledge must be imparted through prenatal clinics, and particularly those serving low-income vulnerable populations.”

This statement contradicts the finding regarding sources of information regarding Zika. Study participants reported that their main sources of information include the Media/Internet (65%), physician nurse (32.3%), and clinic (20%). Therefore, even if clinics provided “factual knowledge” the populations being served will not be reached. This recommendation should be removed. If possible replace with recommendations based on public health campaign messaging literature.

Line 368-369

“Our study indicated that single mothers are at a higher risk of not having such critical knowledge.”

Data to support this statement is not presented. None of the p-values in table 2 are significant for the questions when examined by “couple status.”

Page 21

Lines 380-383

Authors argue that because over half of mothers reported an unintended pregnancy, they could have travelled to high risk “counties” (I think authors meant countries). This is a big leap given that they only obtained travel information from a limited number of mothers. I would remove text starting with “Nearly…” in line 380 ending in “…sex.” In line 383.

Lines 384-385

Authors argue that mothers did not care to postpone a pregnancy despite the Zika outbreak. However, the study was conducted in a community that was at low risk for Zika at the time. Therefore, this statement should be removed or the authors need to provide stronger argument that this community was at high risk. Underserved pregnant Latinas are not at high risk for Zika due to their ethnicity, rather, the community outbreak status determines what the level of risk is for pregnant women.

Page 22

Lines 400-402

Remove statement “These groups are viewed as hard-to-reach populations, not only because of language barrier and lower trust in media, but also because they report relatively low use of various media channels.”

This statement has no reference. See previous statements from line 95-97. “Hard to reach” is not a justifiable reason to study a community/population. Additionally, internet media reaches this community very effectively.

Lines 402-404

Please expand the implications suggested.

Lines 411-415

Authors argue that HCPs need to understand the sources of information patients get information from, which this study identified as internet being the primary source for information on Zika among pregnant Latinas. However, authors then argue that accessibility is an issue, which contradicts study findings. Lastly, the “Internet should be considered a substitute” statement is quite strong, given that the internet can refer to a multitude of sources for information. Please clarify and consider providing a recommendation that HCPs can suggest to patients as a good source of information regarding Zika.

Page 23

Line 420

Remove “targeted” and insert “patient”

Lines 423-424

Suggest how HCPs can accomplish offering culturally appropriate Zika risk counseling. Expand this recommendation.

Lines 432-433

See previous statements from line 95-97 and 400-402. “Hard to reach” is not a justifiable reason to study a community/population. Additionally, internet media reaches this community very effectively.

Line 433

Please provide a more definitive statement rather than stating the “sample is ‘expected’ to be inclusive of 85% of pregnant Latinas in the county.”

Page 24

Lines 460-465

Authors offer a variety of recommendations to provide patient education, however, these suggestions can be weaved throughout the discussion narrative by providing references to show effectiveness of these efforts. Authors can strengthen these implications by bolstering the discussion section.

Page 25

Lines 486-487

See previous statements from line 95-97, 400-402, and 486-487. “Hard to reach” is not a justifiable reason to study a community/population.

Page 26

Lines 489-490

I would remove this statement. A quick search yielded several studies. Of note, see study by Winneg et. al., Risk Analysis, Vol. 38, No. 12, 2018.

Lines 492-496

I recommend removing implications for COVID-19 until there is enough scientific evidence to explain how COVID-19 puts pregnant women at risk. However, the implications for dissemination of information is still applicable.

Page 27

Lines 512-513

HCPs consists of a variety of health professionals. Suggesting that HCPs need training and preparation is ok, but I suggest that the conclusion provide more explicitly who HCPs can collaborate with to make this feasible.

Reviewer #3: I did not participate in the first review, however it appears that the authors fully addressed the first reviewers concerns. This is a god paper and a relevant topic. I would like to call out a few minor issues to address before being published:

1) Line 48, there is a small typo - "and" should be changed to "who"

2) Line 50/ 51 - denoted traveled to "by" March 2016. Should this be "in"? Is this time period of a single year or several years. Can you clarify? Also, what is meant by "passengers"? Passengers sounds like they arrived by train, plane or car however I don't see anything indicating the mode of transportation.

3) I recommend being consistent throughout the paper with calling it ZIKV. There are many instances where it goes back and forth between the abbreviation and Zika.

4) Lines 92-93 - where are the other two clinics located? It would be nice to know a little more about the other two clinics. Also, there is a small typo on line 93, the word "the" needs to be inserted before Texas-Mexico border.

5) Line 120 - typo in Cameron County (not Cameroon)

6) Line 140 - typo, should be "they" waited or "waiting"

7) Line 167 - typo, should read "at" least 8 weeks

8) Lines 258-259 contains a redundant statement

9) Line 338 - typo, need to add the word "they" to became pregnant

10) Line 382, typo, should be "countries", not counties

This research is commendable in that it identifies important gaps in knowledge transfer in particularly vulnerable women. Comparing between FQHCs and private clinics is key as it illustrates how each have areas to be improved.

7. PLOS authors have the option to publish the peer review history of their article (what does this mean?). If published, this will include your full peer review and any attached files.

Reviewer #2: No

Reviewer #3: **Yes: **Paula Stigler Granados

---

## [Author Response · Author response to Decision Letter 1]

30 Sep 2020

We have attached a separate document with responses to each reviewers' comments.

---

## [Decision Letter · Decision Letter 2]

16 Oct 2020

PONE-D-19-24828R2

Communicating the Risk of Contracting Zika Virus to Low Income Underserved Pregnant Latinas: A Clinic-based Study

PLOS ONE

Dear Dr. RAMISETTY-MIKLER,

Thank you for submitting your manuscript to PLOS ONE. After careful consideration, we feel that it has merit but does not fully meet PLOS ONE’s publication criteria as it currently stands. Therefore, we invite you to submit a revised version of the manuscript that addresses the points raised during the review process.

In view of the comment provided by Reviewer 2 (see below) we request to consider whether it is possible to describe why your study population is a hard-to-reach group - or whether this group is just underrepresented in previous research.

We look forward to receiving your revised manuscript.

Kind regards,

Florian Fischer

Academic Editor

PLOS ONE

Additional Editor Comments (if provided):

Please check the references once more. E.g. in lines 441-448, the references are not correctly ordered and sometimes you are using brackets and sometimes square brackets.

Reviewers' comments:

Reviewer's Responses to Questions

**Comments to the Author**

1. If the authors have adequately addressed your comments raised in a previous round of review and you feel that this manuscript is now acceptable for publication, you may indicate that here to bypass the “Comments to the Author” section, enter your conflict of interest statement in the “Confidential to Editor” section, and submit your "Accept" recommendation.

Reviewer #2: All comments have been addressed

Reviewer #3: All comments have been addressed

2. Is the manuscript technically sound, and do the data support the conclusions?

Reviewer #2: Yes

Reviewer #3: Yes

3. Has the statistical analysis been performed appropriately and rigorously? 

Reviewer #2: Yes

Reviewer #3: Yes

4. Have the authors made all data underlying the findings in their manuscript fully available?

Reviewer #2: Yes

Reviewer #3: Yes

5. Is the manuscript presented in an intelligible fashion and written in standard English?

Reviewer #2: Yes

Reviewer #3: Yes

6. Review Comments to the Author

Reviewer #2: Thank you for the opportunity to review the author’s revisions on the study titled, “Communicating the risk of contracting Zika virus to low income underserved pregnant Latinas: A clinic-based study.” The article is very much improved. The topic is timely and relevant. It adds to the body of work with ethnically diverse populations in the U.S.

I have one major comment:

Lines 442-444, lines 489-490

Authors attempted to justify the term “hard-to-reach” using a citation that refers to mostly geographical challenges in accessing a community. This citation also describes sampling techniques.

In the context of this study, I disagree with the use of this term. The term “hard-to-reach” in this study actually refers to researchers’ challenges in recruiting ethnically diverse participants. However, the authors explain in their methods that they completed their study in a county with almost 20% Latinx population, utilized clinics that serve this population, and they trained staff to recruit participants. If a researcher has the appropriate staffing and training, these communities are not “hard-to-reach,” which is the case in this study as evidenced by what they report in their methods. I think the more appropriate term for this study is “underrepresented in research” not “hard-to-reach.”

I realize that this term is a nuanced by fields. I leave it to the editor to decide if this term is appropriate for the journal - hence my minor revision recommendation.

Reviewer #3: The authors addressed all of the concerns in the paper. I appreciate these efforts as this is a very understudied population that is difficult to reach.

7. PLOS authors have the option to publish the peer review history of their article (what does this mean?). If published, this will include your full peer review and any attached files.

Reviewer #2: No

Reviewer #3: **Yes: **Paula Stigler Granados

---

## [Author Response · Author response to Decision Letter 2]

16 Oct 2020

T

hank you for the reviewers’ comments. We have made requested changes

Additional Editor Comments:

Please check the references once more. E.g. in lines 441-448, the references are not correctly ordered and sometimes you are using brackets and sometimes square brackets. 

All the brackets have now been corrected to square brackets. The references were inserted using Mendeley and I am not sure why some a few of the brackets are different.

Reviewers' comments and response:

In view of the comment provided by Reviewer 2 (see below) 

Reviewer #2: 

I have one major comment:

Lines 442-444, lines 489-490 

We have now deleted the term "hard-to-reach" and changed it to underrepresented in research.

---

## [Editor Report · Decision Letter 3]

20 Oct 2020

Communicating the Risk of Contracting Zika Virus to Low Income Underserved Pregnant Latinas: A Clinic-based Study

PONE-D-19-24828R3

Dear Dr. RAMISETTY-MIKLER,

We’re pleased to inform you that your manuscript has been judged scientifically suitable for publication and will be formally accepted for publication once it meets all outstanding technical requirements.

Kind regards,

Florian Fischer

Academic Editor

PLOS ONE
---

## [Editor Report · Acceptance letter]

4 Nov 2020

PONE-D-19-24828R3 

Communicating the risk of contracting Zika virus tolow income underserved pregnant Latinas: A clinic-based study  

Dear Dr. RAMISETTY-MIKLER:

I'm pleased to inform you that your manuscript has been deemed suitable for publication in PLOS ONE. Congratulations! Your manuscript is now with our production department. 

Kind regards, 

on behalf of

Dr. Florian Fischer 

Academic Editor

PLOS ONE